# Study on Physical Properties, Rheological Properties, and Self-Healing Properties of Epoxy Resin Modified Asphalt

**Jiasheng Li, Yaoyang Zhu \* and Jianying Yu**

State Key Laboratory of Silicate Materials for Architectures, Wuhan University of Technology, Luoshi Road 122, Wuhan 430070, China
* Correspondence: yaoyang@whut.edu.cn

**Abstract:** To investigate the effects of epoxy resin at low content on the physical properties, rheological properties, and self-healing properties of asphalt, epoxy asphalts with epoxy resin contents of 2%, 5%, 10%, and 20% were prepared. The distribution of epoxy asphalt (EA) in epoxy resin (ER) was quantitatively studied by fluorescence microscopy (FM) to investigate the feasibility of the preparation process. The glass transition temperature of epoxy asphalt was quantitatively analyzed by the differential thermal analyzer (DSC). The physical properties of epoxy asphalt were characterized by penetration test, ductility test, and softening point test. The rheological properties of epoxy asphalt were analyzed by the dynamic shear rheometer (DSR) to evaluate the self-healing properties of epoxy asphalt. The results show that the epoxy resin could be uniformly distributed in the asphalt, as verified by fluorescence microscopy (FM). With the increase in epoxy resin content, the glass transition temperature of epoxy asphalt gradually decreases, and the epoxy asphalt with 20% content shows the lowest glass transition temperature. At the same time, epoxy resin gives asphalt a higher modulus and high temperature performance, and the penetration and softening point of epoxy asphalt has also been greatly improved. On the contrary, the three-dimensional cross-linked grid structure, which is formed by epoxy resin and curing agent, reduces the rheological properties of epoxy asphalt and increases the elastic components of epoxy asphalt. Although the maltenes diagram still exhibits typical viscoelastic characteristic, the flow behavior index and flow activation energy of epoxy asphalt decreased.

**Keywords:** epoxy asphalt; epoxy resin; physical properties; rheological properties; self-healing properties





## 1. Introduction

Asphalt, which is a by-product of oil production, is widely used in pavement due to its low cost and easy accessibility [1–3]. However, asphalt is a low molecular thermoplastic compound, and the properties of asphalt are highly correlated with temperature [4,5]. High temperature causes the asphalt to flow, resulting in rutting on the pavement; low temperature increases the modulus of asphalt, resulting in brittle cracking of pavement [6]. To change the temperature sensitivity of asphalt, a large number of researchers have developed thermoplastic modified asphalt, such as styrene (SBS) [7,8], waste rubber [9–13], and low-density polyethylene [14] based on the concept of environmental protection [15,16] and durability, which effectively improves the service life of the pavement. However, with the continuous increase in traffic volume, buildings represented by long-span bridges have high requirements on the strength, corrosion resistance, fatigue resistance, and deformation resistance of pavement [17]. As thermoplastic modified asphalt struggles to meet the requirements of bridge deck pavement, thermosetting materials were gradually developed and proved to be more effective [18–22].

Epoxy asphalt is a thermosetting modified asphalt [23–25], which is a modified binder made by mixing epoxy resin and asphalt. Since the epoxy resin and the curing agent

undergo an irreversible cross-linking and curing reaction, the asphalt molecules are fixed by a three-dimensional cross-linked grid structure [26–29]. This curing reaction gives asphalt excellent physical and chemical properties, with good high temperature performance, low temperature crack resistance, fatigue resistance, and durability [30,31]. Therefore, the use of epoxy asphalt for pavements not only helps to improve the service life of pavements but also greatly increases the road value and economic benefits of pavements. However, with the heavy vehicle loads continuously impacting the pavement, it is difficult to ensure that the pavement will not develop cracks with epoxy asphalt as the pavement material [32–35]. To repair cracks, common repair methods mainly use preventive means to prevent water from penetrating the base. However, these preventive measures are not only difficult when providing long-term protection measures for the pavement, but also cause serious traffic congestion, which does not meet the long-term requirements of social development.

Since asphalt can be identified as a self-healing material, cracks can be repaired by heating the pavement [36–38]. Using the self-healing of asphalt to repair the pavement can not only reduce road congestion but also reduce the waste of resources. Therefore, since the self-healing phenomenon of asphalt was first reported, the self-healing of asphalt has been a research hotspot. Garcia [39] first proposed that capillary flow and diffusion are the main mechanisms of self-healing of asphalt binders by studying the self-healing properties of asphalt; he found that there is a threshold temperature for self-healing. He believes that when the self-healing reaches the threshold temperature, the asphalt will transform into a near-Newtonian fluid. Li [40] found that the initial temperature of self-healing of asphalt emulsion was closely related to the fluidity of asphalt emulsion by studying the effect of asphalt emulsion on the self-healing of asphalt. Zhang [41] prepared modified asphalt with different aging degrees for flow characteristics analysis and found that the fluidity of asphalt will decline when it is aged to a certain extent, and it is difficult for asphalt to self-heal. Tang [42] compared the self-healing properties of different asphalt binders and found that different asphalt binders have different rheological properties and threshold temperatures. Therefore, the self-healing performance of asphalt is closely related to the ambient temperature. The higher the ambient temperature, the stronger the fluidity of asphalt and the faster the self-healing rate. Although epoxy resin provides excellent physical and chemical properties for asphalt, it reduces the fluidity of asphalt. Therefore, studying the influence mechanism of different amounts of epoxy resin on the rheological properties of asphalt can quantitatively analyze the influence of epoxy resin on the self-healing properties of asphalt. At the same time, because the epoxy resin is expensive, a low blending amount of epoxy resin can not only improve the mechanical value of the pavement but also reduce the influence of epoxy resin on flow properties.

To fill the research gap, epoxy asphalts containing 2%, 5%, 10%, and 20% were prepared. The physical properties, rheological properties, and self-healing properties of epoxy asphalt with different contents were studied from micro and macro scales. Microscopically, the distribution of epoxy resin in asphalt was studied by fluorescence microscope (FM). The glass transition temperature of epoxy asphalt was studied by the differential thermal analyzer (DSC). Macroscopically, the effects of different content of epoxy resin on the physical properties of asphalt were studied by penetration, softening point, and ductility. The rheological properties and self-healing properties of epoxy asphalt were studied by dynamic shear rheometer (DSR).

## 2. Materials and Methods

### 2.1. Materials

#### 2.1.1. Base Asphalt

The raw materials of epoxy asphalt are mainly divided into epoxy resin and base asphalt. The base asphalt is selected as 90# base asphalt, which is commonly used in pavement engineering. The basic properties of asphalt are shown in Table 1.

**Table 1.** Basic properties of 90# base asphalt.

| Indexes | Experimental Method | Test Data |
|---|---|---|
| Penetration 25 °C (0.1 mm) | T0604 | 93.7 |
| Ductility, 5 cm/min, 15 °C (cm) | T0605 | >150 |
| Softening point (°C) | T0606 | 47.2 |

2.1.2. Epoxy Resin (ER)

The epoxy resin grade is E51(618), which is purchased from Hangzhou Wuhuigang Adhesive Co. The ratio of epoxy resin to curing agent is 1:1, and the basic properties of epoxy resin will be shown in Table 2.

**Table 2.** Basic properties of E51 epoxy resin.

| Test | Experimental Method | Result |
|---|---|---|
| Viscosity (73 °C) (P.s) | ASTM D445 | 17,320 |
| Gardner Color Scale | ASTM D1544 | <1 |
| Water content (%) | ASTM D1744 | <0.5 |
| Proportion (g/eq) | ASTM D1475 | 0.51 |

*2.2. Preparation of Epoxy Asphalt*

To study the modification effect of epoxy resin with different contents on asphalt, the mass ratios of epoxy resin to asphalt were 2%, 5%, 10%, and 20%, respectively. At the same time, considering that epoxy resin is difficult to mix with asphalt, a shear mixer was used for high-speed mixing. The specific preparation process can be divided into the following four steps [43]. Firstly, asphalt and epoxy resin were heated to 135 °C and 80 °C, respectively. Subsequently, the asphalt and epoxy resin were stirred at a shear stirring rate of 1500 rpm for 10 min. After that, the curing agent was added to the mixture, and the heating temperature was 135 °C, the stirring rate was 1000 rpm, and the stirring time was 5 min. Finally, epoxy asphalt was cured at 60 °C for 48 h and marked as 90, 90-2%, 90-5%, 90-10%, and 90-20%, respectively. The production process is shown in Figure 1.

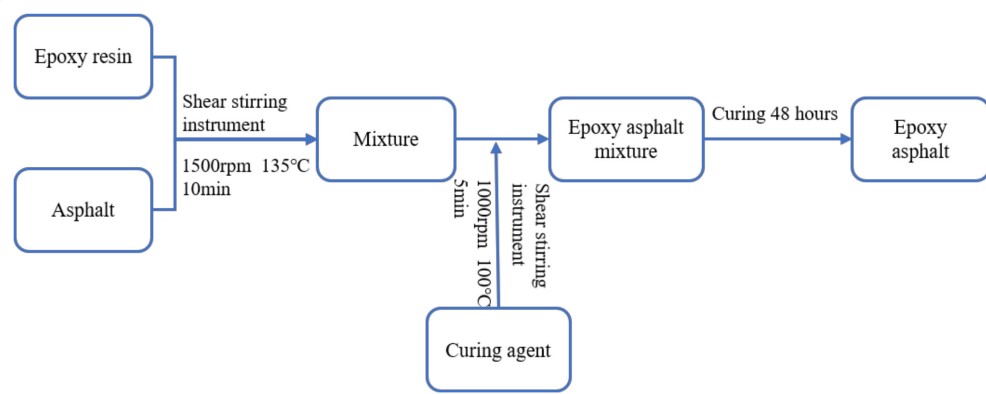

**Figure 1.** Preparation process of epoxy asphalt.

*2.3. Microscopic Test*

2.3.1. Fluorescence Microscopy Test

Fluorescence microscopy (CX23, OLYMPUS, Wuhan, China) was used to observe the distribution of epoxy resin 10 times and 40 times. Due to the different valence bonds of epoxy resin and asphalt, they are luminescent under the fluorescence microscope. Therefore, the distribution of epoxy resin in asphalt can be observed by fluorescence microscopy.

### 2.3.2. Differential Thermal Analyzer Test

To analyze the effect of epoxy resin content on the thermal stability of epoxy asphalt, the glass transition temperature ($T_g$) of epoxy asphalt was determined by DSC, which was used as an index to evaluate the low temperature performance of epoxy asphalt. Based on this, the scanning temperature of DSC is set to −50~10 °C, and the heating rate is 10 °C/min. The test samples were matrix asphalt and four groups of epoxy-modified asphalt, and the glass transition temperature was used as the standard to study the modification effect of epoxy resin on asphalt.

### *2.4. Macroscopic Test*
### 2.4.1. Penetration Test

According to the specification JTG E20-2011, the epoxy asphalt was poured into the mold and cooled at room temperature for 1.5 h. Subsequently, it was placed in a constant temperature water tank at 25 °C for 1.5 h. Finally, the penetration meter was used to repeat the test three times, and the average value of the penetration, taken three times at the same temperature, was taken as the penetration of epoxy asphalt at a certain temperature.

### 2.4.2. Ductility Test

According to the specification JTG E20-2011, the epoxy asphalt was first poured into three molds and cooled at room temperature for 1.5 h. Subsequently, it was placed into the ductility meter at a constant temperature of 10 °C for 1.5 h. After demolding, it was pulled at a speed of 5 ± 0.25 cm/min, and the average value of three groups of specimens fractured was used to evaluate the ductility of epoxy asphalt at room temperature.

### 2.4.3. Softening Point Test

According to the specification JTG E20-2011, two sets of specimens of epoxy asphalt were poured. We added water to the beaker of the softening point tester and placed the specimens into the beaker. Subsequently, we heated the beaker to 5 °C and kept it warm for more than 15 min. Finally, we started the softening point tester and heated the beaker at 5 °C. When the two groups of specimens were dropped to the bottom plate, the average value of the two groups of specimens was taken as the softening point.

### 2.4.4. Dynamic Shear Rheometer Test

DSR was used for frequency scanning tests to analyze the rheological properties and self-healing properties of epoxy asphalt at high temperature. Firstly, epoxy asphalt with different epoxy resin contents was placed between two parallel plates of DSR with a diameter of 25 mm and a spacing of 1 mm. The upper plate was a rotatable plate with a precision motor, and the lower plate was a fixed plate for placing the sample. The frequency sweep test (0.1–100 Hz) was carried out at nine different temperatures (from 34 °C to 82 °C at intervals of 6 °C) for each sample. The high temperature rheological properties of epoxy asphalt were studied by composite viscosity ($\eta$), flow behavior index, and flow activation energy.

To study the viscoelastic properties of epoxy asphalt over a wide range of frequencies, the Williams–Landel–Ferry (WLF) model was used to construct the master curve [43], as shown in Equation (1):

$$\log \alpha_T = \frac{-C_1(T - T_0)}{C_2 + (T - T_0)} \tag{1}$$

where $\alpha_T$ is the apparent displacement factor, $C_1$ and $C_2$ are constants, $T$ is the measured temperature of DSR, $T_0$ is the reference temperature, and the initial temperature of DSR (34 °C) is used as the reference temperature in this paper. The complex modulus and phase angle of the epoxy asphalt are fitted according to the following Equations (2)–(5):

$$f_R = f \times \alpha_T = f \times 10^{\frac{c_2(T - T_R)}{c_2 + (T - T_R)}} \tag{2}$$

$$G^* = G_{\min} + (G_{\max} - G_{\min}) \times \left(1 - \exp\left(-\left(\frac{f_R}{\beta_G}\right)^{\gamma_G}\right)\right) \tag{3}$$

$$\delta^* = \delta_{\min} + (\delta_{\max} - \delta_{\min}) \times \left(1 - \exp\left(-\left(\frac{f_R}{\beta_\delta}\right)^{\gamma_\delta}\right)\right) \tag{4}$$

where $f$ is the frequency of DSR, $G^*$ is the complex modulus, $G_{\max}$ and $G_{\min}$ are the complex modulus at infinity and 0, respectively, $\delta^*$ is the phase angle, $\delta_{\max}$ and $\delta_{\min}$ are the phase angle at infinity and 0, respectively, and $\beta_G$, $\beta_\delta$, $\gamma_G$ and $\gamma_\delta$ are the curve parameters.

To test the rheological properties of epoxy asphalt, the flow behavior factors of epoxy asphalt are fitted by power law relationship [40,41,44], as follows:

$$\eta^* = m|\omega|^{n-1} \tag{5}$$

where $\eta^*$ is the composite viscosity of epoxy asphalt, $m$ and $n$ are the fitting data, and $\omega$ is the frequency of DSR. In theory, epoxy asphalt could be identified as a near-Newtonian fluid with flow characteristics when $n = 0.9 \sim 1$. Therefore, 0.9 is used as the self-healing threshold of epoxy asphalt to characterize the rheological properties of epoxy asphalt.

The corresponding flow activation energy reflects the viscosity-temperature dependence of the specimen [42]. In theory, the higher the flow activation energy, the higher the temperature sensitivity of the specimen. In this paper, the complex viscosity of epoxy asphalt at different temperatures is obtained by the Arrhenius equation, and the flow activation energy of epoxy asphalt is given, as shown in Equation (6):

$$Ln\eta = LnA - \frac{E}{RT} \tag{6}$$

where $\eta$ is viscosity, $E$ is the low activation energy, and $R$ and $T$ are the gas constant and absolute temperature, respectively.

## 3. Results and Discussion

### 3.1. Microscopic Analysis of Epoxy Asphalt

3.1.1. Fluorescence Microscopy Test Result

Figure 2 is the fluorescence microscopy of epoxy asphalt under 10 times mirror and 40 times mirror. Since the epoxy group exhibits a fluorescent state under ultraviolet irradiation, the light spot in the figure represents the epoxy group. From the results of Figure 2, it can be seen that with the increase in epoxy resin content, the epoxy resin groups with the fluorescence effect gradually increased. Among them, the matrix asphalt (a) and (b) are reflected in the absence of epoxy resin, and there is no light spot of an epoxy group in the figure. The light spots of a small amount of epoxy groups appeared in the diagrams of (c) and (d), indicating that the crosslinking curing reaction of the epoxy resin occurred. With the increase in epoxy resin content, a large number of light spots of epoxy groups appear in the (i) and (j) diagrams and are evenly distributed in the matrix asphalt. However, the distribution of light spots in the (e) and (f) plots is not satisfactory, which may be due to the problem of glass film production, resulting in poor observation. Overall, the shear stirrer preparation method can effectively distribute the epoxy resin uniformly in the asphalt to achieve the desired dispersion effect.

3.1.2. Glass Transition Temperature

The glass transition temperature ($T_g$) represents the lowest temperature point at which the macromolecular segment of an amorphous polymer can undergo free movement. In theory, the lower the glass transition temperature ($T_g$) of modified asphalt, the better the mechanical properties of modified asphalt. Figure 3 shows the glass transition temperature ($T_g$) of base asphalt and epoxy asphalt. To obtain the glass transition temperature ($T_g$) of base asphalt and epoxy asphalt, the intersection point of the baseline and the curve is

drawn by the equidistant method, and the temperature at the 1/2 position of the baseline is taken as the glass transition temperature ($T_g$) of base asphalt and epoxy asphalt. At the same time, the affinity of epoxy resin with polar asphaltene groups in asphalt will cause multi-layer adsorption of epoxy resin on the asphalt surface. This adsorption will increase the mobility of non-polar molecular chains in soft asphalt, resulting in a decrease in the glass transition temperature ($T_g$) of asphalt.

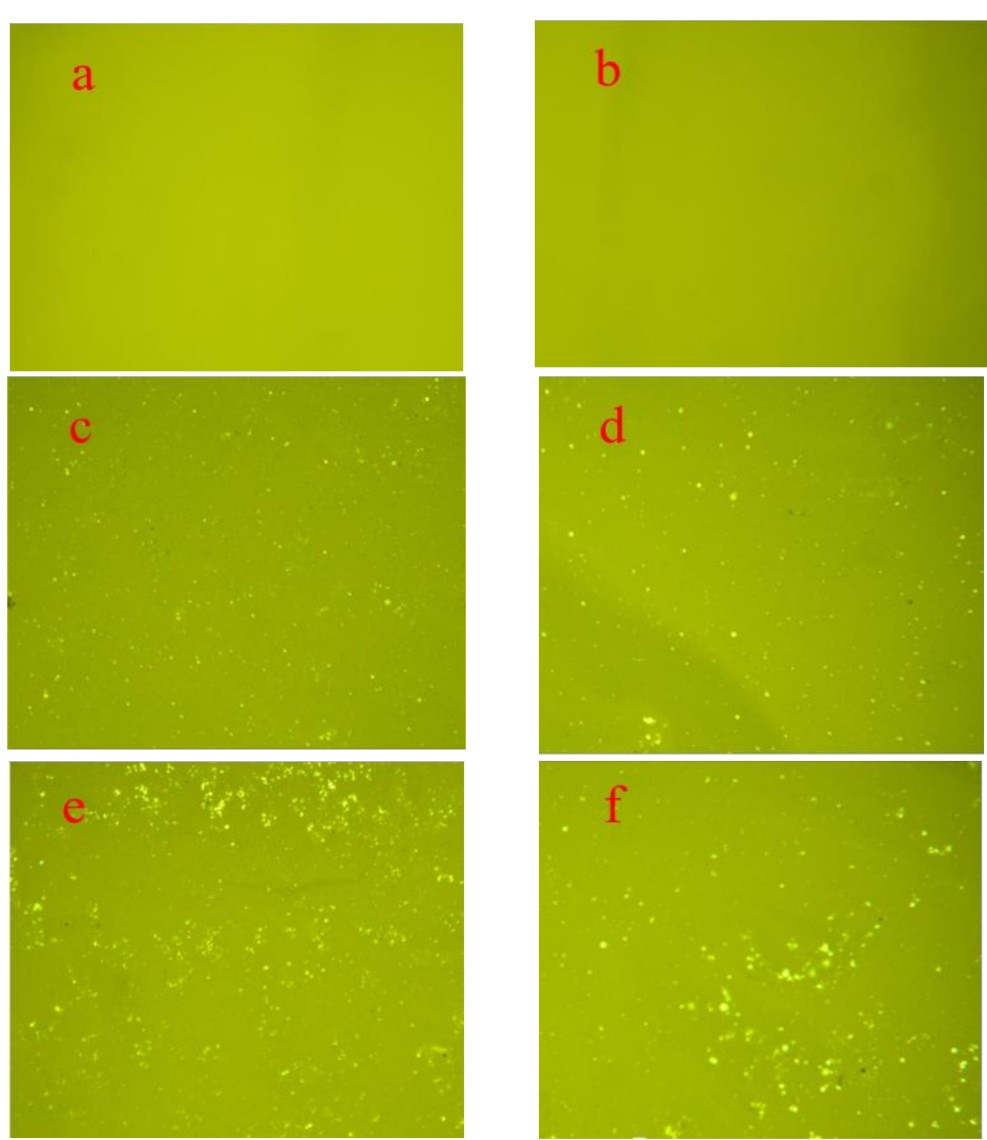

**Figure 2.** *Cont.*

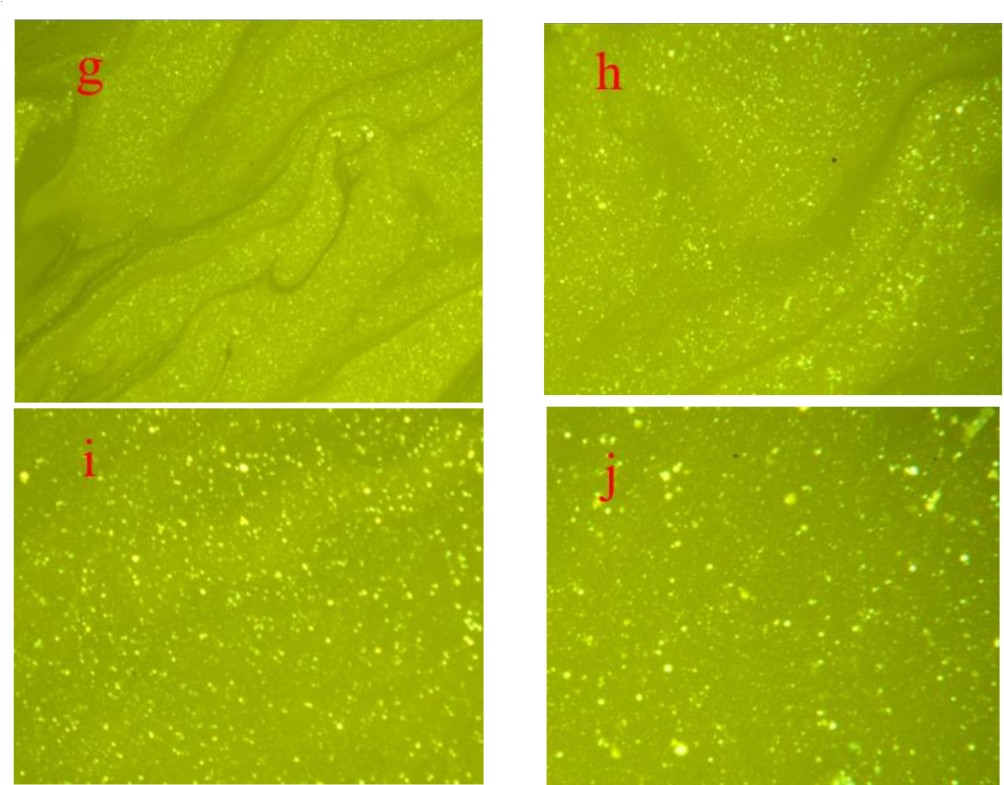

**Figure 2.** Fluorescence microscopy of epoxy asphalt. (**a**) Base asphalt 10×; (**b**) base asphalt 40×; (**c**) 90-2% 10×; (**d**) 90-2% 40×; (**e**) 90-5% 10×; (**f**) 90-5% 40×; (**g**) 90-10% 10×; (**h**) 90-10% 40×; (**i**) 90-20% 10×; (**j**) 90-20% 40×.

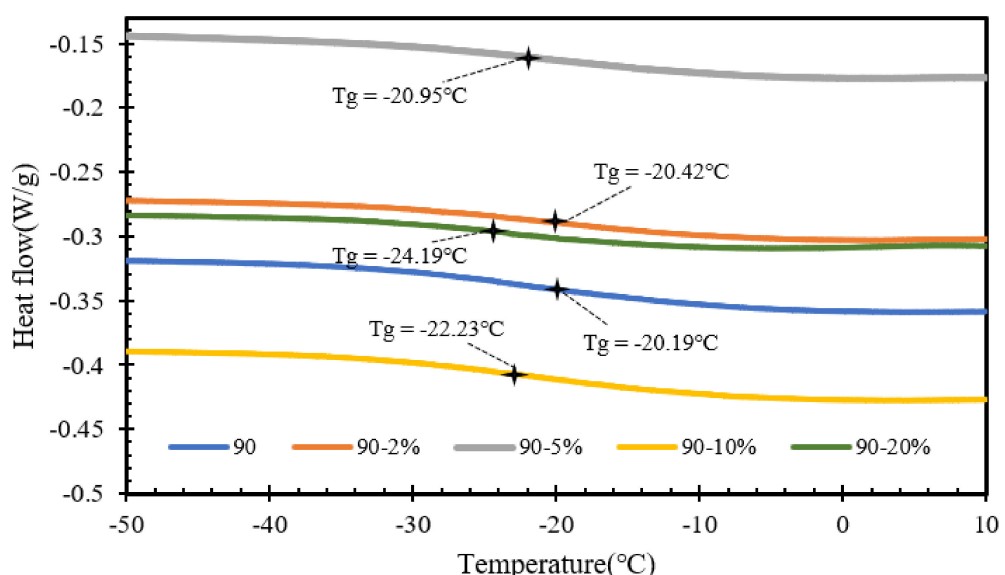

**Figure 3.** DSC curves of base asphalt and epoxy asphalt.

Therefore, the results of Figure 4 show that epoxy resin reduces the glass transition temperature ($T_g$) of base asphalt. The glass transition temperatures ($T_g$) of base asphalt and epoxy asphalt were −20.19 °C. After adding epoxy resin, the glass transition temperature of epoxy asphalt decreased to −20.42 °C, −20.95 °C, −22.23 °C, and −24.19 °C, respectively. From the DSC results, it can be found that the glass transition temperature of epoxy asphalt decreases gradually with the increase in epoxy resin doping, due to the multilayer adsorption of epoxy resin on the asphalt surface, which is consistent with the theoretical

conjecture. Therefore, this phenomenon shows that epoxy asphalt can effectively improve the mechanical properties of asphalt, and with the increase in epoxy resin content, the degree of improvement will be more significant.

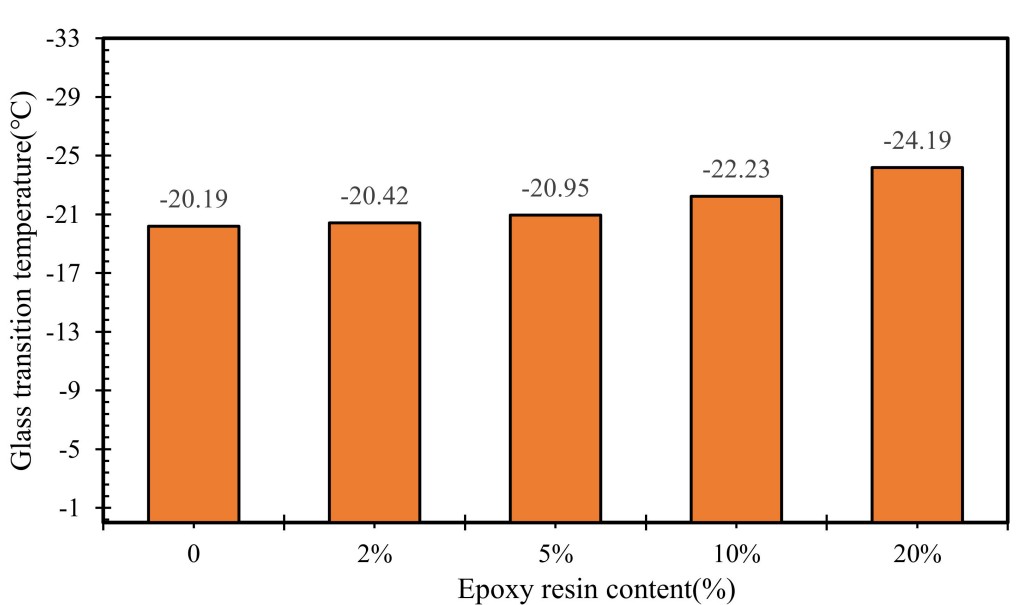

**Figure 4.** Transition temperature of asphalt glass under different epoxy resin content.

### 3.2. Physical Properties of Epoxy Asphalt

Figure 5 shows the physical properties of epoxy asphalt, including the softening point, ductility, and penetration of epoxy asphalt. The softening point, ductility, and penetration of the base asphalt without epoxy resin are 46.3 °C, 150 cm, and 93 mm, respectively. The softening point, ductility, and penetration of epoxy asphalt change from 48.2 °C, 58 cm, and 90 mm to 57.4 °C, 41 cm, and 61 mm with the increase in epoxy resin content, respectively. The reason for the difference is that epoxy resin gives modified asphalt a higher modulus and better high temperature performance. The higher modulus increases the hardness of epoxy asphalt, thereby reducing the ductility and penetration, and greatly increasing the softening point temperature of epoxy asphalt. It can be seen that the addition of epoxy resin can effectively improve the high temperature performance and physical properties of asphalt. However, too high content of epoxy resin will greatly increase the stiffness of modified asphalt, resulting in better fatigue resistance of asphalt [45]. Therefore, it is necessary to investigate the optimal content of epoxy resin.

### 3.3. Rheological Properties of Epoxy Asphalt

### 3.3.1. Complex Modulus and Phase Angle

Figure 6 is the master curve of base asphalt and epoxy asphalt. With the increase in epoxy asphalt content, the modulus of epoxy asphalt increases gradually, while the phase angle decreases gradually. The sample with 20% epoxy resin has the largest complex modulus and the lowest phase angle. This shows that the three-dimensional cross-linked grid structure generated can effectively fix the light components of asphalt and provide more elastic components for asphalt. Compared with base asphalt, epoxy asphalt has better elastic properties and better anti-rutting performance under the action of heavy vehicle loads. This feature is of great significance to improve the service life of the pavement.

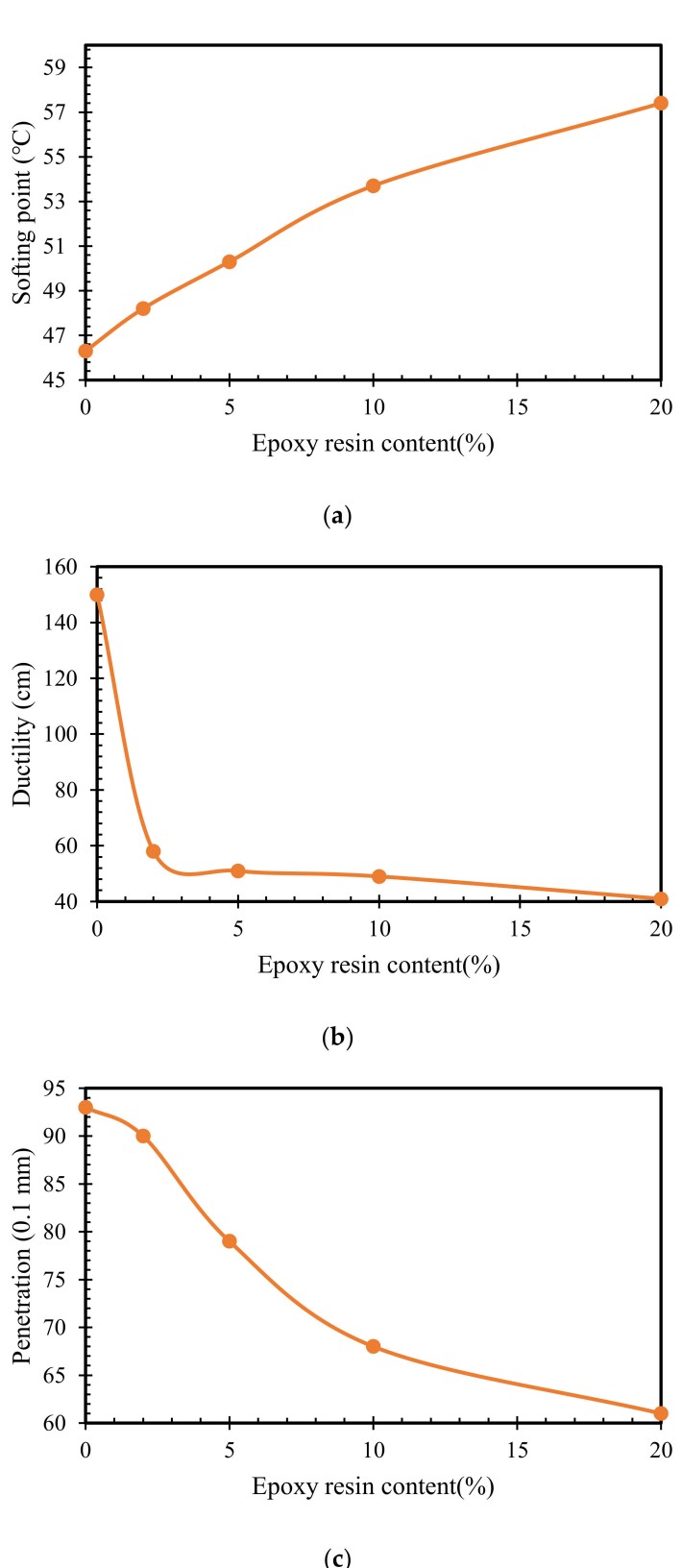

**Figure 5.** Physical properties of epoxy resin. (**a**) Softening point, (**b**) Ductility, (**c**) Penetration.

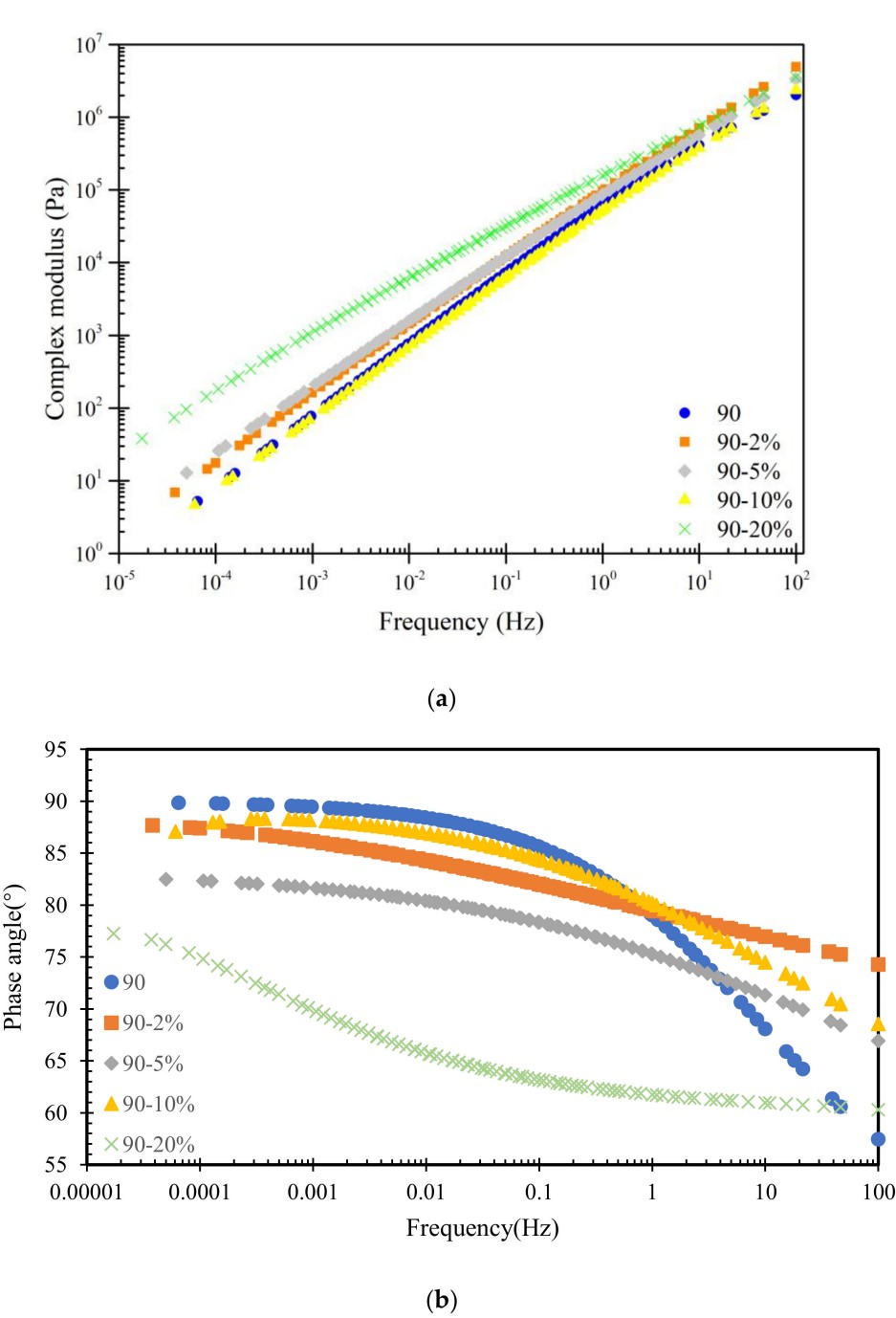

(**a**)

(**b**)

**Figure 6.** Master curve of epoxy asphalt. (**a**) Complex modulus; (**b**) phase angle.

### 3.3.2. Complex Viscosity

Figure 7 shows the frequency–complex viscosity relationship of epoxy asphalt. It can be seen from the diagram that 90-20% epoxy asphalt has the highest complex viscosity. When the temperature is lower than 64 °C, the complex viscosity of asphalt decreases linearly with the increase in frequency. With the increase in temperature, the complex viscosity of epoxy asphalt is close to a straight line. This phenomenon indicates that the complex viscosity of epoxy asphalt no longer changes with the increase in frequency. In general, with the increase in epoxy resin content, the complex viscosity of epoxy asphalt will gradually increase. This phenomenon indicates that the content of epoxy resin effectively improves the high temperature deformation resistance of asphalt.

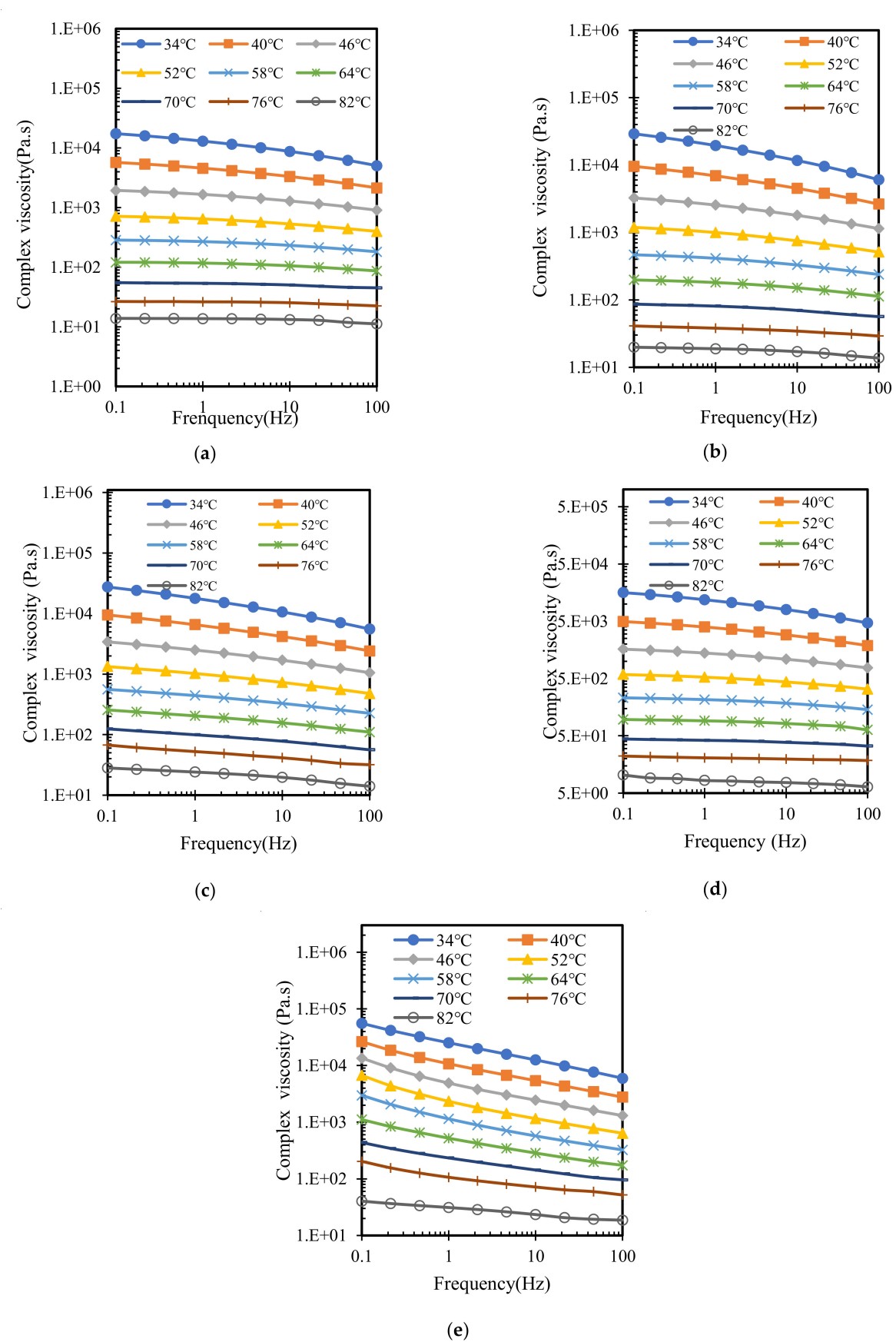

**Figure 7.** Viscosity–frequency curves of the epoxy asphalt at different temperatures. (**a**) 90; (**b**) 90-2%; (**c**) 90-5%; (**d**) 90-10%; (**e**) 90-20%.

### 3.4. Self-Healing Properties of Epoxy Asphalt

3.4.1. Flow Behavior Factors

Table 3 is the fitting result of complex viscosity. In theory, the larger the value of *a* in the fitting formula, the higher the temperature sensitivity of epoxy asphalt. According to the results of Table 2, 90-20% epoxy asphalt has the highest temperature sensitivity, and the flow behavior index decreases with the increase in epoxy resin content. Among them, the range of the flow behavior factors of base asphalt is 0.823–0.977, and the ranges of the flow behavior index of 90-2%, 90-5%, 90-10%, and 90-20% epoxy asphalt are 0.774–0.949, 0.77–0.901, 0.825–0.941, and 0.683–0.883, respectively.

**Table 3.** Fitting results of flow behavior factors.

| Material | Temperature | Fitting Formula $(y = ax^b)$ A Value | b Value | $R^2$ | Flow Behavior Index |
|---|---|---|---|---|---|
| 90 | 34 | 12,535 | −0.177 | 0.9814 | 0.823 |
| | 40 | 4427.3 | −0.141 | 0.9794 | 0.859 |
| | 46 | 1608.5 | −0.110 | 0.9699 | 0.890 |
| | 52 | 625.5 | −0.085 | 0.9566 | 0.915 |
| | 58 | 260.7 | −0.065 | 0.9271 | 0.935 |
| | 64 | 114.3 | −0.048 | 0.8975 | 0.952 |
| | 70 | 52.8 | −0.030 | 0.9126 | 0.970 |
| | 76 | 25.9 | −0.023 | 0.8203 | 0.977 |
| | 82 | 13.535 | −0.027 | 0.7414 | 0.973 |
| 90-2% | 34 | 18,758 | −0.226 | 0.9886 | 0.774 |
| | 40 | 6687.8 | −0.187 | 0.9878 | 0.813 |
| | 46 | 2461.7 | −0.152 | 0.9815 | 0.848 |
| | 52 | 963.54 | −0.123 | 0.9726 | 0.877 |
| | 58 | 399.46 | −0.098 | 0.959 | 0.902 |
| | 64 | 175.23 | −0.081 | 0.9435 | 0.919 |
| | 70 | 78.7 | −0.063 | 0.9457 | 0.937 |
| | 76 | 37.6 | −0.048 | 0.9686 | 0.952 |
| | 82 | 18.5 | −0.051 | 0.908 | 0.949 |
| 90-5% | 34 | 17,311 | −0.23 | 0.9924 | 0.77 |
| | 40 | 6371.7 | −0.197 | 0.9931 | 0.803 |
| | 46 | 2431.7 | −0.169 | 0.9913 | 0.831 |
| | 52 | 995.6 | −0.148 | 0.9896 | 0.852 |
| | 58 | 432.01 | −0.131 | 0.9893 | 0.869 |
| | 64 | 200.93 | −0.121 | 0.9899 | 0.879 |
| | 70 | 98.4 | −0.113 | 0.9931 | 0.887 |
| | 76 | 52.4 | −0.109 | 0.9979 | 0.891 |
| | 82 | 23.58 | −0.099 | 0.969 | 0.901 |
| 90-10% | 34 | 11,426 | −0.175 | 0.9814 | 0.825 |
| | 40 | 3833.9 | −0.139 | 0.9797 | 0.861 |
| | 46 | 1347.9 | −0.108 | 0.9718 | 0.892 |
| | 52 | 512.35 | −0.084 | 0.9599 | 0.916 |
| | 58 | 208.31 | −0.064 | 0.9375 | 0.936 |
| | 64 | 89.4 | −0.054 | 0.8926 | 0.946 |
| | 70 | 41.3 | −0.039 | 0.9429 | 0.961 |
| | 76 | 20.78 | −0.024 | 0.9936 | 0.976 |
| | 82 | 8.6 | −0.059 | 0.9614 | 0.941 |
| 90-20% | 34 | 25,809 | −0.317 | 0.9982 | 0.683 |
| | 40 | 11,382 | −0.317 | 0.9885 | 0.683 |
| | 46 | 5371.5 | −0.327 | 0.9766 | 0.673 |
| | 52 | 2584.2 | −0.329 | 0.9715 | 0.671 |
| | 58 | 1237.5 | −0.314 | 0.9809 | 0.686 |
| | 64 | 546.11 | −0.267 | 0.9902 | 0.733 |

**Table 3.** *Cont.*

| Material | Temperature | Fitting Formula (y = ax^b) | | R² | Flow Behavior Index |
|---|---|---|---|---|---|
| | | A | b | | |
| | | Value | Value | | |
| | 70 | 243.04 | −0.219 | 0.9893 | 0.781 |
| | 76 | 114.7 | −0.188 | 0.9638 | 0.812 |
| | 82 | 30.9 | −0.117 | 0.9961 | 0.883 |

Figure 8 shows the flow behavior index of epoxy asphalt at different temperatures. Combined with the results in Table 3, it can be found that the flow properties of asphalt decreased significantly after adding epoxy resin. This is because the three-dimensional cross-linked grid structure by epoxy resin fixes the light component of asphalt, which makes it difficult for epoxy asphalt to produce an obvious flow. The flow behavior index of base asphalt, 90-2%, and 90-10% at 58 °C reached 0.9, indicating that the three epoxy asphalts all exhibited near-Newtonian liquid states and reached the self-healing threshold. However, it should be noted that the flow behavior factor of the 90-10% specimens was similar to that of the matrix asphalt and showed a sharp decrease at 76 °C. This may be due to the curing reaction of the epoxy resin with the hardener again at high temperature, which led to the decrease in the flow behavior factor at 76 °C for the 90-10% specimens. At the same time, the physical properties of epoxy asphalt were found to increase with the increase in epoxy resin parameters, and the elasticity and viscosity of epoxy asphalt were significantly increased. Due to the increase in the elasticity and viscosity of the binder, the ability of the pavement to resist heavy vehicle loads can be effectively improved. This phenomenon shows that 90-2% and 90-10% have better self-healing properties. Due to the poor flow performance of 90-5% and 90-20%, and the high temperature sensitivity of the two epoxy asphalts, they are not conducive to the stability of the pavement.

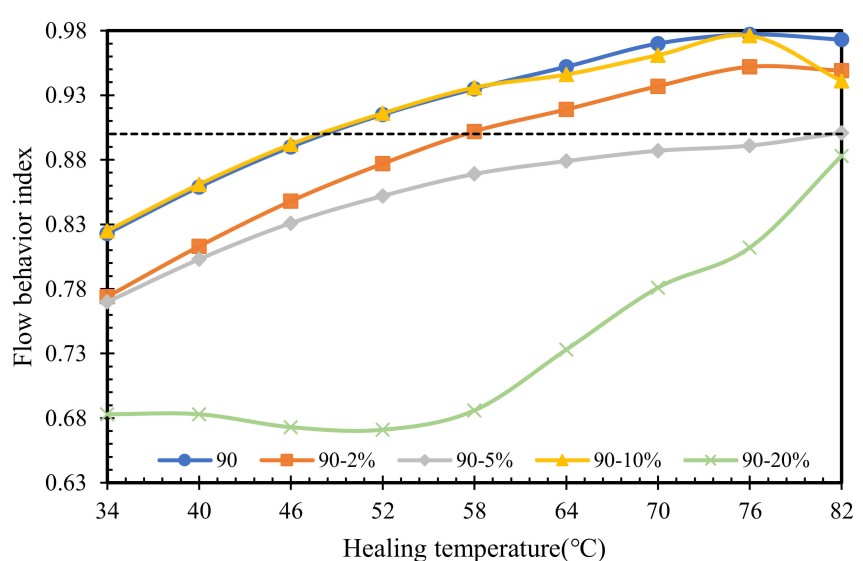

**Figure 8.** Flow behavior factors of base asphalt and epoxy asphalt.

3.4.2. Flow Activation Energy

Figures 9 and 10 is the Arrhenius diagram of maltenes and flow activation energy, and Table 4 is the fitting curve of base asphalt and epoxy asphalt. The product of the slope of the fitting curve and the gas constant is the flow activation energy of epoxy asphalt. Among the samples, the flow activation energy of 90-5% and 90-20% epoxy asphalt was significantly smaller than other specimens. The 90-2% and 90-10% epoxy asphalt showed better flow

properties, which was consistent with the results of the flow behavior index. At the same time, it can be seen that the flow properties of 90-10% epoxy asphalt decline sharply when it is at 76 °C. This is due to the curing of epoxy resin at high temperatures, which reduces the flow properties of epoxy asphalt. In general, the addition of epoxy resin will give asphalt higher viscosity but will reduce the fluidity of epoxy asphalt. Therefore, under the condition of ensuring the strength of asphalt, the maximum content of epoxy resin should be controlled to ensure the self-healing performance of epoxy asphalt. Combining the effects of epoxy resin on the physical properties, rheological properties, and self-healing properties of asphalt, 90-2% specimens and 90-5% specimens possess better rheological properties and can provide excellent self-healing properties for pavements. However, the low admixture of epoxy resin resulted in less elastic components and poorer deformation resistance. In contrast, 90-20% specimens have the highest elastic component, but the rheological properties are too low to achieve the flow of asphalt at high temperatures. Therefore, the addition of 10% epoxy asphalt can not only improve the physical properties of asphalt but also ensure the rheological properties and self-healing ability of asphalt.

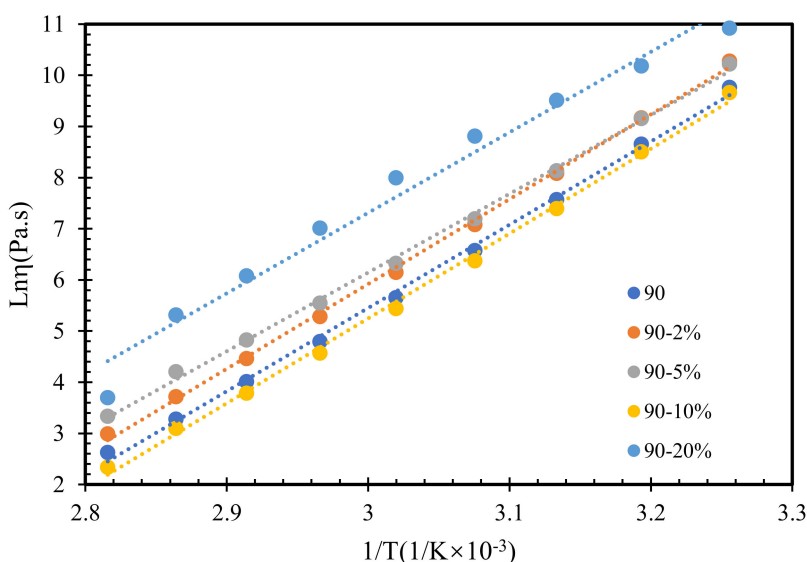

**Figure 9.** Arrhenius diagram of maltenes in base asphalt and epoxy asphalt.

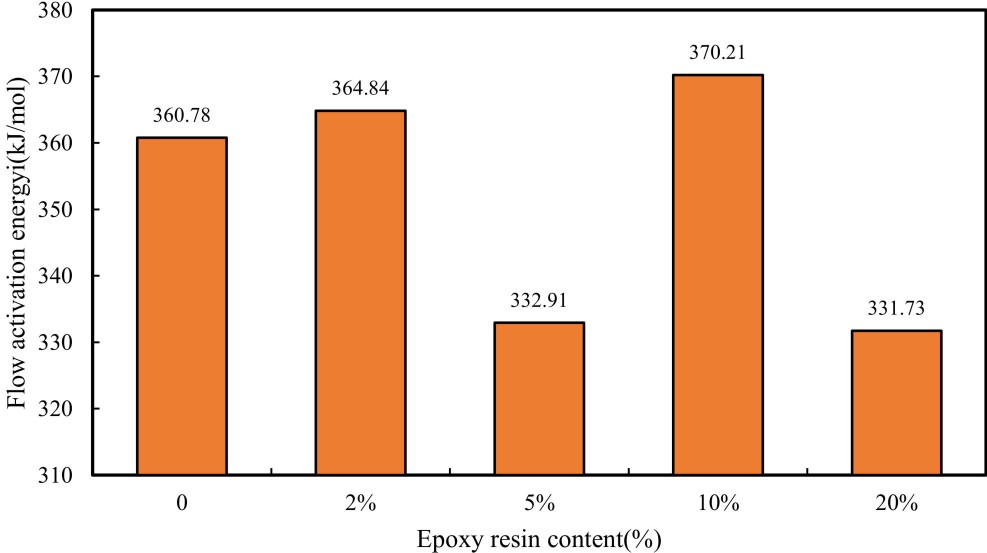

**Figure 10.** Flow activation energy of base asphalt and epoxy asphalt.

**Table 4.** Flow activation energy fitting data.

| Material | Fitting Formula (y = ax + b) | | $R^2$ | Flow Activation Energy |
|---|---|---|---|---|
| | a | b | | |
| | Value | Value | | |
| 90 | 16.281 | −43.394 | 0.9979 | 360.7777 |
| 90-2% | 16.584 | −43.883 | 0.9988 | 364.8433 |
| 90-5% | 15.395 | −40.042 | 0.9980 | 332.9092 |
| 90-10% | 16.592 | −44.529 | 0.9976 | 370.2141 |
| 90-20% | 15.737 | −39.900 | 0.9771 | 331.7286 |

## 4. Conclusions

To fill the gap in the self-healing research of epoxy asphalt, the effects of different contents of epoxy resin on the physical properties and self-healing properties of epoxy asphalt were studied. The physical properties of epoxy asphalt were measured by penetration, ductility, softening point, and glass transition temperature tests. The rheological properties and self-healing properties of epoxy asphalt were characterized by flow behavior factor and flow activation energy. According to the above results, the following can be concluded:

(1) Through the observation of fluorescence microscopy, it is found that the distribution of epoxy functional groups in asphalt is dense and uniform, indicating that epoxy resin can be uniformly dispersed in asphalt under the action of 135 °C and high-speed shearing, which meets the mixing requirements of asphalt in actual construction.

(2) With the increase in epoxy resin content, the glass transition temperature of asphalt decreased from −20.19 °C to −24.19 °C. This may be because the multilayer adsorption of epoxy resin and asphalt will increase the mobility of non-polar molecular chains in soft asphalt, thereby reducing the glass transition temperature of epoxy asphalt. This property can effectively improve the low temperature mechanical properties of epoxy asphalt.

(3) With the increase in epoxy resin content, the physical properties represented by softening point, ductility, and penetration are improved. This may be because the three-dimensional cross-linked grid structure generated by epoxy resin and the curing agent gives asphalt better stability, thereby improving the physical properties of asphalt.

(4) The addition of epoxy resin reduces the rheological properties of asphalt, resulting in a decline in the self-healing properties of epoxy asphalt. The flow behavior index of 90-2% and 90-5% epoxy asphalt at 76 °C are 0.976 and 0.952, respectively, which are much larger than the respective 0.891 and 0.812 of 90-2% and 90-5% epoxy asphalt.

(5) Considering the improvement in the physical properties of the epoxy resin and the reduction in the rheological properties of the asphalt, the 10% content of epoxy asphalt can not only have better elasticity and viscosity but can also make the flow behavior factor greater than 0.9, which can ensure the self-healing performance of the asphalt. Therefore, the best epoxy resin content for this study was set at 10%.

This study verified the feasibility of high-temperature self-healing of epoxy asphalt and found that modified asphalt with epoxy resin content higher than 20% is difficult to rely on at high temperatures to achieve self-healing of pavement. However, the self-healing of pavement does not only depend on the flow of asphalt itself, and a better self-healing effect can be achieved by adding capsules. Therefore, this study is only applicable to the self-healing performance of asphalt at high temperatures, and other self-healing means are to be studied in subsequent papers.

**Author Contributions:** J.L.: methodology, investigation, writing—original Draft. Y.Z.: Conceptualization, supervision. J.Y.: conceptualization, validation. All authors have read and agreed to the published version of the manuscript.

**Funding:** The authors gratefully acknowledge the financial support provided by the National Natural Science Foundation of China (51978547), Hubei Science and Technology Innovation Talent and Service Project (International Science and Technology Cooperation) (2022EHB006) and Technological Innovation Major Project of Hubei Province (2019AEE023), Fujian Provincial Science and Technology Planning Project (2022H0027).

**Institutional Review Board Statement:** Not applicable.

**Informed Consent Statement:** Not applicable.

**Data Availability Statement:** The data presented in this study are available on request from the corresponding author.

**Conflicts of Interest:** The authors declare that they have no conflict of interest.

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
