# Peer review of "Study on Physical Properties, Rheological Properties, and Self-Healing Properties of Epoxy Resin Modified Asphalt"

_sustainability, doi:10.3390/su15086889_

Round 1

Reviewer 1 Report

This paper experimentally investigates the effects of mixing epoxy with asphalt at various content. Overall the article is interesting but requires some improvements. 

Line 12:

Has the glass transition temperature been measured by DSC (differential scanning calorimetry) or DTA (differential thermal analyzer)?

Line 14

Please make sure you use abbreviation correctly and efficiently.

Line 16

“With the increase of epoxy resin content, the glass transition temperature of epoxy asphalt gradually decreases, showing good mechanical properties” – this statement is vague, please revise.

Reference 37 is not in correct format.

What did the authors mean by prescribed values in Table 1 and 2?

Line 96 - spelling

Is it curing 48 (text line 108) or 24 hours (schematic)?

Line 120

How many samples were tested in DSC?

Line 139 Subsection missing

Line 201

Looking at Figures 2e and f, the dispersion does not look “ideal”. I would suggest changing the wording.

Line 202 What is a Tg of epoxy itself? Can you please include it in Table2?

Line 215 – Can you please include stdev?

Line 218 – 220 this statement is vague.

Line 232 - Any reference to support this statement?

Figure 8 – Looking at the Figure 8, it seems that flow behaviour of reference asphalt and EA 90-10% is very similar. Any comments on the behaviour that compared to, for example, 90-2%? Is it related to the mixing? 

What is the optimal epoxy content considering all properties?

Reviewer 2 Report

This manuscript reports the study on the properties of epoxy resin-modified asphalt. I have a few comments as follows:

1. The paper could benefit from more discussion about the potential practical applications and implications of the study, such as how the findings could inform the design and construction of more durable and sustainable pavements.

2. Fig.2: why does the 90-10% sample show some dark areas in the g and f?

3. Fig.4: how many samples were tested for each bar, if more than one, please add an error bar.

4. Fig.6: it’s very confusing, please use different line styles for different y-axes.

Reviewer 3 Report

The manuscript needs to be modified, based on the following comments:

1.       The abstract requires to be revised, covering the basic subject matter and the research outcomes.

2.       The literature should be enhanced, since some recent and important conclusions are missing:

https://doi.org/10.1155/2021/5513338

https://doi.org/10.1016/j.jclepro.2022.135030

http://dx.doi.org/10.1002/app.33948

3.       The research gap, objective and methodology must be improved, preferably under separate headings.

4.       The significance importance, novelty and limitations of the work must be included, preferably under separate headings.

5.       The conclusions should focus on the primary research findings.

6.       It is not clear why the epoxy resin content is limited to 20% only, and why not the higher values.

7.       Figure 6, 8, 9: The legends are not clear.

8.       Caption of Figure 7 is not clear.   

9.       The curve fitting data were presented in Tabular form (Table 3). It is not clear why only one type of curve were chosen while the values of R2 are low in quite a few cases. A graphical representation of curve fittings would also be required.

Round 2

Reviewer 2 Report

Fig.6: please use different line styles for the sample of 90-20%

Reviewer 3 Report

Satisfactory. 

Author Response

Thank you for your valuable comments on this article, because of your hard work to have the highlights of this article. Thanks again for your comments!